# Importance of Communication Skills Training and Meaning Centered Psychotherapy Concepts among Patients and Caregivers Coping with Advanced Cancer

**DOI:** 10.3390/ijerph20054458

**Published:** 2023-03-02

**Authors:** Normarie Torres-Blasco, Lianel Rosario-Ramos, Maria Elena Navedo, Cristina Peña-Vargas, Rosario Costas-Muñiz, Eida Castro-Figueroa

**Affiliations:** 1School of Behavioral and Brain Sciences, Ponce Health Sciences University, Ponce, PR 00716, USA; 2Ponce Research Institute, Ponce Health Sciences University, Ponce, PR 00716, USA; 3School of Medicine, University of Connecticut, Storrs, CT 06269, USA; 4Department of Psychiatry & Behavioral Sciences, Memorial Sloan-Kettering Cancer Center, New York, NY 10065, USA

**Keywords:** communication, meaning, advanced cancer, patients, caregiver, dyads, Latino

## Abstract

Latinos are more likely to be diagnosed with advanced cancer and have specific existential and communication needs. Concepts within Meaning-Centered Psychotherapy (MCP) interventions and Communications Skills Training (CST) assist patients in attending to these needs. However, Latino-tailored MCP interventions have yet to be adapted for advanced cancer patients and caregivers. A cross-sectional survey was administered to Latino advanced cancer patients and caregivers where participants rated the importance of the goals and concepts of MCP and CST. Fifty-seven (*n* = 57) Latino advanced cancer patients and fifty-seven (*n* = 57) caregivers completed the survey. Most participants rated MCP concepts as extremely important, ranging from 73.75% to 95.5%. Additionally, 86.8% favored finding meaning in their life after a cancer diagnosis. Participants (80.7%) also selected the concept of finding and maintaining hope to cope with their cancer diagnosis. Finally, participants found CST concepts and skills acceptable, ranging from 81.6% to 91.2%. Results indicate the acceptability of Meaning-Centered Therapy and Communication Skills Training among Latino advanced cancer patients and caregivers coping with advanced cancer. These results will inform the topics to be discussed in a culturally adapted psychosocial intervention for advanced cancer patients and their informal caregivers.

## 1. Introduction

Foreign-born Latinos, from countries such as Cuba, Puerto Rico, Mexico, and Central and South America, are more likely to be diagnosed with cancer at an advanced stage when compared to non-Latino whites [1,2,3]. In addition, foreign-born Latinos have specific existential [4] and communication needs [5,6]. An advanced cancer diagnosis can cause physical, emotional, psychosocial, and existential stress not only for the patient but also for the caregiver [7,8,9,10]. Cancer as a significant stressor has been treated with several psychotherapeutic interventions designed to address this existential suffering and communication need. Specifically, Meaning-Centered Psychotherapy (MCP) and Communication Skills Training (CST) have shown an effect by targeting the specific psycho-spiritual needs of patients with advanced cancer and enhancing a sense of meaning, peace, and purpose as they face an advanced diagnosis [11,12,13], while CST targets communication skills with patients and caregivers coping with cancer [14].

William Breitbart developed Meaning-Centered Psychotherapy (MCP) as an intervention to address the existential distress often experienced by patients with advanced cancer [13]. What differentiates MCP from other types of psychotherapy is its direct approach to identifying sources of meaning in the patient’s life through a set list of strategies grounded in the work of Viktor Frankl [13]. A clinical trial comparing Individual Meaning-Centered Psychotherapy (IMCP), Supportive Psychotherapy (SP), and Enhanced Usual Care (EUC), or standard care, showed IMCP had significant treatment effects compared to EUC and some modest differences when compared to SP [13]. Patients with advanced cancer are not the only ones who could benefit from MCP. Informal caregivers are also at risk of suffering distress from anxiety, depression, and existential concerns, including “guilt, issues with role changes, sense of identity, and responsibility to the self [15,16]”. A caregiver-focused MCP intervention addresses existential burdens [15,16] and has been found to be feasible and acceptable [17].

Communication between patients and their caregivers is crucial after a diagnosis of advanced cancer, as themes regarding the patient’s values and end-of-life care may surface [18], leading to issues involving death [19], which is still taboo in Hispanic/Latino society [20] and distressing for patients. Given that Hispanics/Latinos are a heterogeneous culture, there are many reasons why death is a taboo subject: fear of expediting the process [21], denial [18], religious matters [21], and sociocultural factors [22]. Skillfully navigating this initial conversation requires a high dexterity in communication on behalf of the provider [23,24]. Some recent studies have focused on communication coaching for patients before appointments with providers [25,26,27]. However, a lesser-studied element is the communication between the patient and their family caregiver. Patient-family communication is an integral part of adapting to the new diagnosis, as family members may take on new roles as informal caregivers and patients adjust to a newly uncertain future. Because dysfunctional communication can be a source of distress for both members of this unique dyad, recent studies have focused on communication skills between patient and caregiver [28,29], and interventions that address communication are being developed [14].

A meta-analytic review has shown that culturally adapted treatments tailored for a specific cultural group are four times more effective than interventions provided to participants from a variety of cultural backgrounds, and those conducted in Latino participants’ native language are twice as effective as interventions conducted in English [30]. Though several psychotherapeutic interventions are designed for advanced cancer patients [11,12,13,31,32,33,34,35,36], only one has been adapted for Latino patients. Nevertheless, interventions have yet to be explicitly adapted for Latino patients and caregivers.

Literature underscores the importance and impact of MCP and communication for advanced cancer patients and their caregivers. Moreover, it highlights the need for culturally adapted interventions. The team used a quantitative approach with patients and caregivers coping with advanced cancer to identify the accepted concepts of Meaning-Centered Psychotherapy and Communications Skills Training. This paper aims to evaluate the importance of MCP concepts and communication concepts among Latino patients and caregivers coping with cancer. The results of this study will be used to inform the topics to be discussed in the psychosocial intervention for advanced cancer patients and their informal caregivers.

## 2. Materials and Methods

Meaning-Centered Psychotherapy is grounded in the work of Dr. Breitbart and aims to target the specific psycho-spiritual needs of patients with advanced cancer [11]. Its primary goal is to help patients enhance a sense of meaning, peace, and purpose as they approach the end of life. The intervention focuses on meaningful concepts such as: maintaining hope, making sense of the cancer experience, having a purpose in life, reflecting on their heritage, having a purpose in life after a cancer diagnosis, changing their attitude, and being responsible for themselves and others after the cancer diagnosis. Moreover, the intervention addresses experiential sources of meaning, such as love, humor, and beauty.

Using the Communication Skills Training approach [14], the team hypothesizes that the MCP intervention will be enhanced by/including taught coping skills. The coping skills training approach was adapted for non-spousal patients’ caregivers by eliminating spousal terms (e.g., taking care of your partner–spouse) and using general caregiving terms (e.g., taking care of your significant other). The concepts related to CST involve learning how to share thoughts about cancer, express feelings regarding a cancer diagnosis, learn strategies to accept others’ perspectives, and acquire communication strategies to accept and validate others [14].

Participants for this study comprised patients and caregivers who were recruited as dyads from an oncology clinic in the southern area of Puerto Rico between October 2020 and September 2021. The Ponce Research Institute Institutional Review Board (IRB) and Ethical Committee approved all the study procedures. An IRB-approved introductory letter to familiarize potential participants with the study. Patients’ inclusion criteria included: (1) patients with solid stage III or IV tumors, (2) age 21 or older, and (3) self-reported Latino. Eligible family caregivers included those who were: (1) a caregiver with a family member diagnosed with solid stage III or IV tumors referred by the advanced cancer patient, (2) age 21 or older, and (3) self-reported Latino. Patients’ exclusion criteria included: (1) diagnosed with a major disabling medical or psychiatric condition, (2) unable to understand the consent procedure, or (3) too ill to participate, reported by the patient and determined by the PI’s judgment. After completing the screening process, those eligible and interested were consented and scheduled to complete the questionnaire. Following informed consent, patients, and family caregivers (FCs) were assigned a subject number and administered the survey and self-report assessment to evaluate the patients’ and FCs’ perspectives and psychosocial needs.

The cross-sectional survey in Spanish included rating the importance of the goals and concepts of MCP and CST. In addition, the survey included general demographic questions (age, education, and gender) and a series of standardized scales described in the protocol paper [37]. Participants were given USD 15 as compensation for their time and effort.

All analyses were conducted using IBM SPSS Statistics 21. The database was checked for coding errors and missing data using descriptive statistics. The analyses included descriptive statistics and frequency analysis for the survey to rate the importance of the goals and concepts of MCP and CST. The study was properly powered to use the findings in this formative work. The G Power statistical program [38] was used to determine the sample size. The study had a power of 0.80 (*p* < 0.05) to detect a medium-sized effect (Cohen’s d = 0.50) [38]. Based on the analysis, the team recruited a sample of 114 participants (57 advanced cancer patients; 57 family caregivers).

## 3. Results

Fifty-seven (*n* = 57) Latino cancer patients with stages III (38.5%) and IV (61.4%) cancer participated. Most patients (57.9%) and caregivers (71.9%) were married. On average, patients were 63 and caregivers were 56 years old. Most patients were male (57.9%), and most caregivers were female (67.9%). The predominant cancer diagnoses were cervical (17.5%), breast (14.0%), and prostate (14.0%). The sociodemographic and diagnostic characteristics are included in Table 1.

### 3.1. Meaning Centered Psychotherapy Concepts: Dyads

When asked about MCP concepts, the majority of participants rated the concepts as extremely important, ranging from 73.75 to 95.5%. Participants (95.5%) ranked the concept of their “love for loved ones” as extremely important when coping with a cancer diagnosis. Participants (91.2%) valued the concept of “maintaining hope” as extremely important. Many participants (89.5%) ranked the concept of “being responsible for themselves after cancer diagnosis” as extremely important. Most participants (89.5%) rated the concept of their “love for life” extremely important when coping with a cancer diagnosis. Participants (87.6%) ranked the concept of “finding beauty in music, nature, and other life experience” as extremely important. Regarding the concept of the “care of others after a cancer diagnosis”, 86% of participants rated it as extremely important. Most participants (81.6%) ranked the concept of “understanding their life’s purpose after a cancer diagnosis” as extremely important. Additional concepts were ranked as extremely important: ”reflecting or thinking about their changes after a cancer diagnosis” (79.8%), “changing or adjusting their attitude when circumstances are out of their control” (77.25), “creating meaning in life or thinking about their purpose” (78.1%), “reflecting on heritage and thinking about their life’s contributions” (75.4%), and “making sense of the cancer experience” (73.7%) (see Table 2 for more details). Most participants (86.8%) chose the concept of “finding meaning in their life after a cancer diagnosis”. Moreover, 80.7% of participants selected the concept of “finding and maintain hope to cope with their cancer diagnosis” (see Table 3).

### 3.2. Meaning Centered Psychotherapy Concepts: Patients

When asked about MCP concepts, most patients rated the concepts as extremely important, ranging from 75.4% to 94.7%. Patients (94.7%) ranked the concept of their “love for loved ones” as extremely important when coping with a cancer diagnosis. Patients (93%) assessed the concept of “maintaining hope” as extremely important. Many patients (91.2%) ranked the concept of “persevering a sense of humor has helped them cope with their cancer diagnosis” as extremely important. Most patients (89.5%) rated the concepts of their “love for life”, “being responsible for themselves after cancer diagnosis”, and “finding beauty in music, nature, and other life experience” as extremely important when coping with a cancer diagnosis Regarding the concept of the “care of others after a cancer diagnosis”, 82.4%% of patients rated it as extremely important. Additional concepts were ranked as extremely important:” create meaning in life or think about their purpose in life” (78.9%), “changing or adjusting their attitude when circumstances are out of their control” (78.9%), “understand their life’s purpose after being diagnosed with cancer” (77.2%), “making sense of the cancer experience” (77.2%), “reflect or think about how they have changed after a cancer diagnosis” (75.4%), and “reflect on their heritage or thinking about what they have contributed with their life” (75.4%) (see Table 4 for more details). Most patients (84.2%) chose the concept of “Find meaning in life after a cancer diagnosis”. Moreover, 80.7% of patients selected “finding and maintaining hope to cope with their cancer diagnosis” (see Table 5).

### 3.3. Meaning Centered Psychotherapy Concepts: Caregivers

When asked about MCP concepts, the majority of caregivers ranked the concepts as extremely important, ranging from 70.2 to 96.4%. Caregivers (96.4%) rated the concept of their “love for loved ones” as extremely important when coping with a cancer diagnosis. Caregivers (89.5%) also rated the concepts of “maintaining hope”, “being responsible for themselves after a cancer diagnosis”, “love for life has helped them cope after a cancer diagnosis”, and “taking care of others after a cancer diagnosis” as extremely important. Most caregivers (86%) ranked the concept of “understanding their life’s purpose after a cancer diagnosis” as extremely important. Caregivers (84.2%) ranked “finding beauty in music, nature, and other life experience” as extremely important. Regarding the concept of “reflect or think about how they have changed after a cancer diagnosis”, 84.2% of caregivers rated it as extremely important. Many caregivers (83.9%) rated “preserving a good sense of humor” as extremely important when coping with a cancer diagnosis. Additional concepts were ranked as extremely important: “creating meaning in life or thinking about their purpose” (77.2%), “changing or adjusting their attitude when circumstances are out of their control” (75.4%), “reflecting on heritage and thinking about their life’s contributions” (75.4%), and “making sense of the cancer experience” (70.2%) (see Table 6 for more details). Most caregivers (89.5%) chose the concept of “finding meaning in their life after a cancer diagnosis”. Moreover, 80.7% of caregivers selected “finding and maintaining hope to cope with their cancer diagnosis” (see Table 7).

### 3.4. Communication Skills Training Concepts:Dyads

When asked about communication skills training concepts and skills, most participants wanted to learn more, ranging from 81.6% to 91.2%. Most participants (91.2%) chose that they would like to acquire problem-solving skills, and 90.4% favored the concept of “wanting to learn that they worry about each other”. A large portion (89.5%) favored the concept of “learning ways to show they are accompanying each other in the process”. Participants (86.6%) also favored the concept of “learning to talk about their thoughts regarding cancer”. Eighty-six percent (86%) of participants favored the concept of “acquiring communication strategies to accept and validate others”. Furthermore, 85.8% of participants chose the concept of “wanting to express their feelings about cancer” and 85.1% selected the concept of “acquiring communication skills to accept other people’s feelings”. Many participants (82.5%) favored the concept of “learning communication strategies to accept other people’s perspectives”. Finally, 81.6% of participants selected the concept of “reviewing their life and considering their heritage” (see Table 8 for more details).

### 3.5. Communication Skills Training Concepts: Patients

When asked about communication skills training concepts and skills, most patients wanted to learn more, ranging from 80.7% to 91.2%. Most patients (91.2%) selected that they would like to acquire problem-solving skills. A large portion (89.5%) favored the concepts of “wanting to learn that they worry about each other”, “learning ways to show they are accompanying each other in the process”, “learning to talk about their thoughts regarding cancer”, and “wanting to express their feelings about cancer”. Patients (87.7%) also favored the concepts of “acquiring communication strategies to accept and validate others” and “acquiring communication skills to accept other people’s feelings”. Furthermore, 84.2% of patients chose the concept of “learning communication strategies to accept other people’s perspectives” Finally, 80.7% of patients selected the concept of “reviewing life and considering their heritage” (see Table 9 for more details).

### 3.6. Communication Skills Training Concepts: Caregivers

When asked about communication skills training concepts and skills, most caregivers wanted to learn more, ranging from 80.7% to 91.2%. Most caregivers (91.2%) chose that they would like to acquire problem-solving skills and favored the concept of “wanting to learn that they worry about each other”. A large portion (89.5%) favored the concept of “learning ways to show they are accompanying each other in the process”. Caregivers (87.7%) also favored “learning to talk about their thoughts regarding cancer”. Additionally, caregivers (84.2%) selected “acquiring communication strategies to accept and validate others”. Moreover, 82.5% of caregivers chose the concepts of “acquiring communication skills to accept other people’s feelings” and “reviewing their life and considering their heritage”. Many (82.1%) favored the strategy of “wanting to express their feelings about cancer”. Finally, 80.7% of caregivers selected the concept of “learning communication strategies to accept other people’s perspective” (see Table 10 for more details).

## 4. Discussion

When patient and caregiver dyads were asked about the concepts of MCP and CST, the majority of participants favorably rated all of the concepts. The acceptance of MCP concepts ranged from 73.75% to 95.5%, while CST ranged from 81.6% to 91.2%. Comparable results were seen in the adaptation of MCP for a Latino population, where patients expressed a need to integrate communication skills as well as accepted MCP concepts in the process of adapting to their cancer diagnosis [37,39]. Some of the many MCP concepts included finding meaning in family and loved ones, maintaining hope, taking responsibility to care for oneself, finding meaning in life after a diagnosis, maintaining a love for life, and preserving a sense of humor. Moreover, the literature acknowledges the efficacy of interventions designed to improve dyadic communication among cancer patients and caregivers [40]. However, studies with Latino patient-caregiver dyads are lacking. A portion of CST concepts includes having problem-solving skills, worrying about each other, demonstrating companionship through the journey, learning to talk about a cancer diagnosis, and acquiring communication strategies.

Results indicate that participants favored love for their loved ones to cope with their diagnosis, which is consistent with studies that underscore how many patients lean on family for support as a coping mechanism during a cancer diagnosis [41]. Additionally, family is an important value to Latinos [42], which could explain why many participants consider it important to take care of others after a cancer diagnosis. Latino patients have reported the desire for assistance in finding hope and meaning in life [43]. Given that participants were also caregivers, many regarded maintaining hope as essential. These results are congruent with literature where caregivers used hope and prayer while caring for a family member with cancer [20]. However, while patients may use hope as a coping mechanism, it can become a difficult topic when discussing end-of-life [19]. The current literature regarding Latino cancer patients and meaning highlights the use of positive reframing and meaning to cope with a cancer diagnosis [41]. Additionally, in the same study, some of the participants integrated the value of life with purpose into their experience with cancer. These results are congruent with the participants’ selection of the concepts of finding meaning, creating meaning, and finding purpose in their life. MCP attempts to assist participants in the search for meaning and purpose through experiential sources of meaning. Moreover, participants favor the discussion of “making sense”, which is seen in advanced cancer patients as an attempt to make sense of and understand the terminality of an advanced cancer diagnosis experience [44].

Many of the participants indicated that reflecting on the changes in one’s life after receiving a cancer diagnosis was important. Even though the MCP concept of change after a diagnosis focuses on general life, hope, and experiences, Latino participants might also reflect on changes attributed to physical changes [45,46], sexuality [46], work [47], or overall quality of life [48]. Regarding responsibility for oneself after a cancer diagnosis, many participants found this to be required within the cancer trajectory. These results are seen in the literature where Latino cancer patients take responsibility for their part in the cancer trajectory [49]. Concerning humor as a coping mechanism, a study with Hispanic male cancer survivors yielded how the survivors used humor as one of many coping mechanisms during their diagnosis and treatment process [50]. Even though our sample includes men, women, patients, and caregivers, most selected humor to cope with their diagnosis.

Some Latinos would rather not discuss the end-of-life stage [18] or death [19]; however, dyads within this study ranked different communication skills as essential. These results could be attributed to integrating cultural factors and values in adapting interventions [51]. The integration of cultural values within interventions has been shown to be successful [52]. For instance, communication interventions aimed at Latinos and their caregivers with a chronic illness (diabetes) yield positive results. Firstly, dyads with good relationships had better care routines, considered the program successful in managing the disease at home, and had better social support [53]. Couples’ CST results underscore the benefits of communication between the advanced cancer patient and partner. Some benefits include: the desire not to be seen as a “patient” and “caregiver,” symptom management, support for a partner, decision making, conflict resolution, and preparation for death [14]. These results are seen within the team’s sample, with participants favoring problem-solving skills and companionship throughout the process. Delivering tailored communication interventions proves to be acceptable and beneficial for the patient and caregiver [54]. Thus, it is highly imperative that caregiver–patient dyads are provided with the necessary skills to discuss thoughts about cancer, express their feelings about cancer, and acquire the necessary communication skills they might need in their daily lives.

## 5. Conclusions

Results within this study show how advanced cancer patients and caregivers favor Meaning-Centered Psychotherapy and Communication Skill training concepts. Existing literature also aids in showing how patients favor these concepts independently and when integrated into an MCP intervention or a communication-based intervention. These results highlight the importance of integrating both patient and caregiver perspectives into the development and application of a culturally adapted psychosocial intervention.

## 6. Limitations

The instrument used in the study, which contained Meaning-Centered Concepts and Communication Skills Training, was developed in a questionnaire manner; therefore, analyses were limited to a descriptive approach. As a result, inferential analyses cannot be measured. If a scale were to be devised, the analysis could explore statistical differences between patients and caregivers, as well as sex, income, and clinical variables.

## Figures and Tables

**Table 1 ijerph-20-04458-t001:** Sociodemographic and clinical characteristics.

Characteristics	Participants, *n* (%)	
	Patients (*n* = 57)	Caregivers (*n* = 57)
Age	63.38 (12.2)	56.22 (14.56)
Sex	Data	Data ^1^
Male	33 (57.9)	18 (32.1)
Female	24 (42.1)	38 (67.9)
Marital Status		
Married	33 (57.9)	41 (71.9)
Single	10 (17.5)	-
Partnered	4 (7.0)	2 (3.5)
Divorced	4 (7.0)	2 (3.5)
Widowed	6 (10.5)	2 (3.5)
Education level		
Never went to school	-	2 (3.6)
Did not finish elementary school	5 (8.8)	1 (1.8)
Elementary school	7 (12.3)	1 (1.8)
Middle school	2(3.5)	4 (7.3)
High school	14 (24.6)	13 (23.6)
Technical or associate degree	12 (21.1)	16 (29.1)
College degree	14 (24.6)	12 (21.8)
Post-college/graduate school	3 (5.3)	6 (10.9)
Employment status		
Employed	8 (14)	13 (22.8)
Employed part-time	-	2 (3.5)
Retired	33 (57.9)	21 (36.8)
Student	1 (1.8)	2 (3.5)
Unemployed	7 (12.3)	10 (17.5)
Disabled	6 (10.5)	4 (7)
Other	2 (3.5)	5 (8.8)
Income		
Less than USD 25,000	36 (63.2)	30 (62.5)
USD 25,001–50,000	14 (24.9)	14 (29.2)
USD 50,001–75,000	-	3 (6.3)
More than USD 75,001	2 (3.8)	1 (1.8)
Health Insurance		
Medicare	26 (45.6)	17 (29.8)
Private	6 (10.5)	16 (28.1)
Public insurance	14 (24.6)	20 (35.1)
Medicare and public insurance	10 (17.5)	3 (5.3)
Medicare and Private	1 (1.8)	1 (1.8)
Diagnosis		
Breast	8 (14.0)	-
Cervical	10 (17.5)	-
Prostate	8 (14.0)	-
Multiple myeloma	7 (12.3)	-
Colon	5 (8.8)	-
Other	19 (33.4)	-
Cancer stage		
III-A	22 (38.5)	-
III-B	7 (12.3)	-
IV	35 (61.4)	-

**Table 2 ijerph-20-04458-t002:** Meaning-centered psychotherapy concepts: Dyads.

Meaning-Centered Therapy Concepts	Extremely Important	Quite Important	Moderately Important	Less Important	Not Important
I think that the love I have for my loved ones has helped me cope with my diagnosis	107 (95.5%)	4 (3.5%)	-	-	1 (0.9%)
Maintaining hope	104 (91.2%)	3 (2.6%)	5 (4.4%)	1 (0.9%)	1 (0.9%)
Being responsible for myself after my cancer diagnosis	102 (89.5%)	4 (3.5%)	3 (2.6%)	1 (0.9%)	4 (3.5%)
I think that the love I have for life has helped me cope with my cancer diagnosis	102 (89.5%)	5 (4.4%)	1 (0.9%)	-	6 (5.3%)
I think that preserving a good sense of humor has helped me cope with my cancer diagnosis	99 (87.6%)	7 (6.2%)	3 (2.7%)	-	4 (3.5%)
I think that finding the beauty in music, nature, and other life experiences has helped me cope with my cancer diagnosis	99 (86.8%)	6 (5.3%)	5 (4.4%)	-	4 (3.5%)
Taking care of others after my cancer diagnosis	98 (86%)	3 (2.6%)	6 (5.3%)	-	7 (6.1%)
Understand my life’s purpose after being diagnosed with cancer	93 (81.6%)	1 (0.9%)	6 (5.3%)	1 (0.9%)	13 (11.4%)
Reflect or think about how I have changed after my cancer diagnosis	91 (79.8%)	6 (5.3%)	4 (3.5%)	1 (0.9%)	12 (10.5%)
Create meaning in my life or think about my purpose in life	89 (78.1%)	8 (7.0%)	4 (3.5%)	5 (4.4%)	8 (7.0%)
Change or adjust my attitude when circumstances are out of my control	88 (77.2%)	8 (7.0%)	8 (7.0%)	-	10 (8.8%)
Reflect on my heritage or thinking about what I have contributed with my life	86 (75.4%)	5 (4.4%)	4 (3.5%)	4 (3.5%)	15 (13.2%)
Making sense of the cancer experience	84 (73.7%)	7 (6.1%)	4 (3.5%)	3 (2.6%)	14 (12.3%)

**Table 3 ijerph-20-04458-t003:** Meaning-centered psychotherapy concepts: Dyads.

Meaning-Centered Therapy Concepts	Yes	Maybe	Not at All
Find meaning in life after my cancer diagnosis	99 (86.8%)	2 (1.8%)	13 (11.4%)
Find and maintain hope to cope with cancer diagnosis	92 (80.7%)	5 (4.4%)	17 (14.9%)

**Table 4 ijerph-20-04458-t004:** Meaning-centered psychotherapy concepts: Patients.

Meaning-Centered Therapy Concepts	Extremely Important	Quite Important	Moderately Important	Less Important	Not Important
I think that the love I have for my loved ones has helped me cope with my diagnosis	54 (94.7%)	2 (3.5%)	-	-	1 (1.8%)
Maintaining hope	53 (93%)	3 (5.3%)	1 (1.8%)	-	-
I think that preserving a good sense of humor has helped me cope with my cancer diagnosis	52 (91.2%)	3 (5.3%)	1 (1.8%)	-	(1.8%)
I think that the love I have for life has helped me cope with my cancer diagnosis	52 (89.5%)	2 (3.5%)	1 (1.85)	-	3 (5.3%)
Being responsible for myself after my cancer diagnosis	51 (89.5%)	1 (1.8%)	2 (3.5%)	1 (1.8%)	2 (3.5%)
I think that finding the beauty in music, nature, and other life experiences has helped me cope with my cancer diagnosis	51 (89.5%)	2 (3.5%)	2 (3.5%)	-	2 (3.5%)
Taking care of others after my cancer diagnosis	47 (82.4%)	1 (1.8%)	5 (8.8%)	-	4 (7%)
Create meaning in my life or think about my purpose in life	45 (78.9%)	3 (5.3%)	3 (5.3%)	2 (3.5%)	4 (7%)
Change or adjust my attitude when circumstances are out of my control	45 (78.9%)	3 (5.3%)	3 (5.3%)	-	6 (10.5%)
Understand my life’s purpose after being diagnosed with cancer	44 (77.2%)	1 (1.8%)	3 (5.3%)	-	9 (15.8%)
Making sense of the cancer experience	44 (77.2%)	1(1.8%)	3 (5.3%)	1 (1.8%)	7 (12.3%)
Reflect or think about how I have changed after my cancer diagnosis	43 (75.4%)	2 (3.5%)	4 (7%)	-	8 (14.6%)
Reflect on my heritage or thinking about what I have contributed with my life	43 (75.4%)	1 (1.8%)	4 (7%)	2 (3.5%)	7 (12.3%)

**Table 5 ijerph-20-04458-t005:** Meaning-centered psychotherapy concepts: Patients.

Meaning-Centered Therapy Concepts	Yes	Maybe	Not at All
Find meaning in life after my cancer diagnosis	48 (84.2%)	1 (1.8%)	8 (14%)
Find and maintain hope to cope with cancer diagnosis	46 (80.7%)	2 (3.5%)	9 (15.8%)

**Table 6 ijerph-20-04458-t006:** Meaning-centered psychotherapy concepts: Caregivers.

Meaning-Centered Therapy Concepts	Extremely Important	Quite Important	Moderately Important	Less Important	Not Important
I think that the love I have for my loved ones has helped me cope with my diagnosis	53 (96.4%)	2 (3.6%)	-	-	-
Maintaining hope	51 (89.5%)	-	4 (7%)	1 (1.8%)	1 (1.8%)
Being responsible for myself after my cancer diagnosis	51 (89.5%)	3 (5.3%)	1 (1.8%)	-	2 (3.5%)
I think that the love I have for life has helped me cope with my cancer diagnosis	51 (89.5%)	3 (5.3%)	-	-	3 (5.3%)
Taking care of others after my cancer diagnosis	51 (89.5%)	2 (3.5%)	1 (1.8%)	-	3 (5.3%)
Understand my life’s purpose after being diagnosed with cancer	49 (86%)	-	3 (5.3%)	1 (1.8%)	4 (7%)
I think that finding the beauty in music, nature, and other life experiences has helped me cope with my cancer diagnosis	48 (84.2%)	4 (7%)	3 (5.3%)	-	2 (3.5%)
Reflect or think about how I have changed after my cancer diagnosis	48 (84.2%)	4 (7%)	-	1 (1.8%)	4 (7%)
I think that preserving a good sense of humor has helped me cope with my cancer diagnosis	47 (83.9%)	4 (7.1%)	2 (3.6%)	-	3 (5.4%)
Create meaning in my life or think about my purpose in life	44 (77.2%)	5 (8.8%)	1 (1.8%)	3 (5.3%)	4 (7%)
Change or adjust my attitude when circumstances are out of my control	43 (75.4%)	5 (8.8%)	5 (8.8%)	-	4 (7%)
Reflect on my heritage or thinking about what I have contributed with my life	43 (75.4%)	4 (7%)	-	2 (3.5%)	8 (14%)
Making sense of the cancer experience	40 (70.2%)	6 (10.5%)	1 (1.8%)	2 (3.5%)	7 (12.3%)

**Table 7 ijerph-20-04458-t007:** Meaning-centered psychotherapy concepts: Caregivers.

Meaning-Centered Therapy Concepts	Yes	Maybe	Not at All
Find meaning in life after my cancer diagnosis	51 (89.5%)	1 (1.8%)	5 (8.8%)
Find and maintain hope to cope with cancer diagnosis	46 (80.7%)	3 (5.3%)	8 (14%)

**Table 8 ijerph-20-04458-t008:** Communication skills training concepts: Dyads.

Communication Skills Concepts	Yes	Maybe	Not at All
Have problem solving skills	104 (91.2%)	3 (2.6%)	7 (6.1%)
Learn that we worry about each other	103 (90.4%)	3 (2.6%)	8 (7%)
Learn ways to show we are accompanying each other in the process	102 (89.5%)	3 (2.6%)	9 (7.9%)
Learn to talk about my thoughts about cancer	101 (86.6%)	2 (1.8%)	11 (9.6%)
Acquire communication strategies to accept and validate others	98 (86%)	3 (2.6%)	13 (11.4%)
Express my feelings about cancer	97 (85.8%)	2 (1.8%)	14 (12.4%)
Acquire communication strategies to accept other people’s feelings	97 (85.1%)	5 (4.4%)	12 (10.5%)
Learn communication strategies to accept other people’s perspective	94 (82.5%)	4 (3.5%)	16 (14%)
Review my life and think about my heritage and contributions	93 (81.6%)	1 (0.9%)	20 (17.5%)

**Table 9 ijerph-20-04458-t009:** Communication skills training concepts: Patients.

Communication Skills Concepts	Yes	Maybe	Not at All
Have problem solving skills	52 (91.2%)	1 (1.8%)	4 (7%)
Learn that we worry about each other	51 (89.5%)	1 (1.8%)	5 (8.85)
Learn ways to show we are accompanying each other in the process	51 (89.5%)	1 (1.8%)	5 (8.8%)
Learn to talk about my thoughts about cancer	51 (89.5%)	1 (1.8%)	5 (8.8%)
Express my feelings about cancer	51 (89.5%)	-	6 (10.5%)
Acquire communication strategies to accept and validate others	50 (87.7%)	1 (1.8)	6 (10.5%)
Acquire communication strategies to accept other people’s feelings	50 (87.7%)	2 (3.5%)	5 (8.8%)
Learn communication strategies to accept other people’s perspective	48 (84.2%)	2 (3.5%)	7 (12.3%)
Review my life and think about my heritage and contributions	46 (80.7%)	-	11 (19.3%

**Table 10 ijerph-20-04458-t010:** Communication skills training concepts: Caregivers.

Communication Skills Concepts	Yes	Maybe	Not at All
Have problem solving skills	52 (91.2%)	2 (3.5%)	3 (5.3%)
Learn that we worry about each other	52 (91.2%)	2 (3.5%)	3 (5.3%)
Learn ways to show we are accompanying each other in the process	51 (89.5%)	2 (3.5%)	2 (7%)
Learn to talk about my thoughts about cancer	50 (87.7%)	1 (1.8%)	6 (10.5%)
Acquire communication strategies to accept and validate others	48 (84.2%)	2 (3.5%)	7 (12.3%)
Acquire communication strategies to accept other people’s feelings	47 (82.5%)	3 (5.3%)	7 (12.3%)
Review my life and think about my heritage and contributions	47 (82.5%)	1 (1.8%)	9 (15.8%)
Express my feelings about cancer	46 (82.1%)	2 (3.6%)	8 (14.3%)
Learn communication strategies to accept other people’s perspective	46 (80.7%)	2 (3.5%)	9 (15.8%)

## Data Availability

Data may be provided upon request.

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
