# Peer review of "Importance of Communication Skills Training and Meaning Centered Psychotherapy Concepts among Patients and Caregivers Coping with Advanced Cancer"

_ijerph, 2023, doi:10.3390/ijerph20054458_

Round 1

Reviewer 1 Report

It is my honor & pleasure reviewing your important research paper

Foreign-born Latinos, from countries such as Cuba, Puerto Rico, Mexico, and Central and South America, are more likely to be diagnosed with cancer at an advanced stage when compared to non-Latino whites [1].

Citing one article to confirm such an important & vital information is ok, however, I would suggest adding more references to authenticate the information

which is still taboo in Hispanic/Latino 59 society [18]

explain why to your reader that such an issue is a is still taboo in Hispanic/Latino

Some recent studies have focused on communication coaching for patients before appointments 62 with providers [21]

You have mentioned some studies. However, we find one citation at the end of the sentence!

Studies have shown that culturally adapted treatments tailored for a specific cultural group are four times more effective than interventions provided to participants 71 from a variety of cultural backgrounds, and those conducted in Latino participants' native language are twice as effective as interventions conducted in English [24].

You have mentioned some studies. However, we find one citation at the end of the sentence!

reflecting on my heritage…. purpose in life after a cancer diagnosis, changing my attitude, and being responsible for myself and others after the cancer diagnosis (line 93-94)

should be:

reflecting on their heritage, changing their attitude, responsible for themselves

Reflecting on your collected data, I would see a significant chance of statistical difference analysis & correlation analysis. It would be an excellent chance for you investing more in your collected data & producing finding that goes beyond descriptive statistics!

The validity & reliability of your tool have not been reported in your methods section!

Author Response

Greetings, 

Attached are responses to reviewer comments. 

Reviewer 2 Report

Dear authors,

Please check my comments in the attached file.

Author Response

(The authors gave the same response as above.)

Reviewer 3 Report

The study presented is interesting, but I think that the analyses carried out are scarce. They could have been completed with others that provide more information, establishing more concrete and clarifying objectives. For example, it would be interesting to know the differences in the average scores according to sex, both of patients and caregivers, of some of the proposed scales. There may be differences in both CCM and CST. This is feasible with student t-analysis or nonparametric tests, depending on the normality of the sample distribution.

In addition, the instruments could be further developed and indicate reliability analyses of the proposed scales.

These changes would improve the document for future publication.

Author Response

(The authors gave the same response as above.)

Round 2

Reviewer 3 Report

Thank you for adding the complementary analyzes that clarify the question that I raised. I have considered accepting this article despite the fact that I believe that in future studies more complex analyzes should be carried out using scales that allow comparisons of means between groups.